# Influence of High-Pressure Die Casting Parameters on the Cooling Rate and the Structure of EN-AC 46000 Alloy

**DOI:** 10.3390/ma15165702

**Published:** 2022-08-18

**Authors:** Wojciech Kowalczyk, Rafał Dańko, Marcin Górny, Magdalena Kawalec, Andriy Burbelko

**Affiliations:** Faculty of Foundry Engineering, AGH University of Science and Technology, Al. A. Mickiewicza 30, 30-059 Krakow, Poland

**Keywords:** high-pressure die casting, aluminium alloy, casting parameters, microstructure

## Abstract

The paper presents the results of the research on the impact of process parameters of high-pressure, cold-chamber die casting of an industrial casting made of aluminium alloy on the casting properties assessed macroscopically by measuring the casting average density and microscopically through the characteristics of the casting microstructure. The analysis covers the influence of three selected velocity settings of the pressing plunger, which determine the filling time, and three values of the compression pressure setting characteristic of the third phase of the casting process. The cooling and solidification simulations of the casting were performed using the ProCAST software. During the simulation tests, the impact of the filling rate of the alloy into the die cavity on the cooling rate and the alloy solidification path at selected points were determined. The conducted research allowed linking the process parameters with the parameters of the casting structure with different wall thicknesses. Metallographic examinations of the castings were carried out using a light microscopy, SEM, and EDS analysis. The fraction of the phases α(Al), the size of dendritic cells, and the size of silicon particles, in the cross-sections of the castings with wall thickness of 3, 6, and 11 mm, respectively, were determined.

## 1. Introduction

High-pressure die casting (HPDC) is treated by the recipient as a homogeneous, basically autonomous object, the assessment of which takes into account the positive fulfilment of certain features defined in the context of intended use and conditions of the use of castings [1,2,3,4]. Aluminium-silicon alloys, in most cases EN-AC 46000, offer the excellent ability to cast components with superior mechanical properties, thin-walled components especially in HPDC. The mechanism of solidification of casting alloys, including aluminium-silicon alloys, may depend on the cooling rate [5,6,7,8,9]. This applies especially to thin-walled castings obtained in the HPDC technology. It is well known that the mechanical properties of casting aluminium alloys are strongly influenced by the local microstructural features that form during solidification, such as grain size, secondary dendritic arm spacing (SDAS), eutectic silicon shape, intermetallic inclusions, and shrinkage [10,11,12,13,14]. The complex process of filling the die cavity in HPDC technology includes three phases: (I) filling the shot sleeve by liquid metal, (II) filling the die cavity at high velocity, (III) applying high pressure (also called intensification pressure) after completion of the die filling when the alloy solidifies. During the production of castings from Al-Si-based alloys using the HPDC method, a strict analysis of the process parameters is necessary [9] This concerns the control of such technological factors as the overheating temperature of the alloy and its chemical composition; modification treatments that affect the primary and eutectic structure; die temperature; the method of filling the metal into the die; cavity venting scheme; and number, shape, and location of filling gates. In the case of HPDC according to [15] process parameters, mainly gate velocity and intensification pressure reduce porosity and increase tensile properties. On the other hand, high values of these parameters increase die erosion. Some practical aspects, such as thin gates and high gate velocities, are critical in avoiding early solidification and are recommended [15]. These conditions affect both the microstructure and the metal turbulence, oxide bi-film formation, and metal flow length, which determine the quality of the castings.

In industrial practice, the basic parameters of the casting process under high pressure (including filling time and compression pressure) are the input data for the correct design of the gating system, and thus the high-pressure die [16]. The same parameters are one of the basic parameters (along with the temperature parameters and the profiles of plunger velocity and pressure changes) which are modified during die filling tests as well as during further optimization of the process in terms of quality (porosity, surface defects, and shape defects) and cost (cycle time, degradation in operation of the high-pressure die).

Suggested ranges of individual HPDC process parameters are available in the literature [17,18,19,20]; however, they are said to be guidelines more for the die designer than for the process engineer parameterizing the operation of the machine (die cavity). At the same time, it is the parameters that are easy to modify that are most often used by process engineers in order to obtain the desired quality or cost effect and, above all, to obtain a stable and homogeneous structure and to eliminate defects of discontinuity and porosity. The technological process is in fact much more complex, and even by obtaining the desired quality effect as described, the economic effect of the process can be significantly deteriorated. An attempt to deepen the understanding of the real effects of changes in these parameters and their impact on the porosity and microstructure of a finished casting could narrow down the wide array of recommended parameters, allowing for their interaction and correlation with other process quantities.

A long-time observation of industrial practice allows us to assume that for a specific type of castings, there must exist such a time range for filling the die cavity (and parameters related to it) that will enable obtaining the best effect of the specific pressure applied, and vice versa, there should be a specific pressure that will give a satisfactory effect, provided that the die is filled in the right time [13,14]. On the same principle, it can be assumed that the change in the time of filling the die cavity has a direct impact on the microstructure of the casting, both in its thin and thicker sections and closer or further away from the gatings.

The castings obtained in the HPDC technology are usually thin-walled ones which are characterized by a high cooling rate and a short solidification time. In the case of HPDC technology and thin-walled castings, the key issue in attaining homogeneous structures and properties free from castings defects is the sensitivity of Al-Si alloys to the cooling rate, especially when the range of the high cooling rates is taken into consideration. The cooling rate (K/s) determines the fragmentation of the dendrites of the α (Al) phase, the type of eutectic (lamellar or fibrous), and the morphology of other phases, including iron ones (sludge). The cooling rate is usually determined when the alloy reaches the equilibrium solidification temperature of the alloy (liquidus temperature). The cooling rate reflects the heat transfer conditions of solidification [9,15] and can be related to the parameters of the castings’ microstructure. The rate of heat dissipation has a direct impact on the number of nuclei of the primary and eutectic phases and on the solidification kinetics. As mentioned, high cooling rates make the production of thin-walled castings difficult. As reported by Jiang [21,22], expandable pattern shell casting can be suitable for high quality and precise aluminium castings with superior mechanical properties. From Reference [21], it also follows that improvement of casting quality through, for example, grain refinement by mechanical vibration is much more difficult in thin-wall castings than in castings with larger wall thicknesses.

As part of the presented work, a casting that was analysed was a component of one of the key systems of a modern internal combustion engine. The objective of the study is to find out the influence of the parameters of the high-pressure die casting process, mainly the pressure as well as the rate and time of filling the die, on the parameters of the casting microstructure in thin and thicker sections located closer to or further from the gatings. The use of numerical modelling allowed simulation of the course of filling the die cavity and of the solidification of alloys in areas with different wall thicknesses. The determined cooling rates and solidification times were linked to the parameters of the HPDC process. These results will extend the knowledge regarding production stability, structure formation, and minimizing generation of casting scrap.

## 2. Description of the Material and Experiments

### 2.1. Material

The base-line material was a secondary AlSi9Cu3(Fe), Alumetal Poland SP. Z.O.O, Kęty, Poland, cast alloy EN AC-46000. The alloy was melted in an 800 kg industrial electric resistance furnace with preheating to a temperature of 1023 ± 5 K (750 ± 5 °C). Alloy composition was determined using a GNR Solaris NF (Analytical Instruments Ltd., Nottingham, UK) optical emission spectrometer on the samples taken directly from the casting (Table 1). All the experimental castings analysed in this work were made from a single pour of the dosing furnace. After the alloy was chemically prepared, it was poured into a batching furnace, where the holding temperature was 973 K (700 °C), for subsequent casting. All castings were made for the tests described below while the machine was running in an automatic cycle with stabilized cycle time (51 s) and process thermal conditions.

A series of numerical simulations were carried out using ProCAST, ESI Group^®^ software (ESI Group, Rungis, France) to determine the local cooling rate of the alloy during the process. For each casting, measurements of piston path, piston speed, and pressure in the pressing cylinder over time were recorded. Based on these real curves, piston motion parameters were created for ProCAST simulations.

The temperature dependencies of the thermophysical parameters of the alloy used, necessary for the simulations, were obtained by thermodynamic calculations. For this aim the CompuTherm LLC Thermodynamic Databases of ProCAST was used [23]. These results were obtained using the Scheil–Guliver model of components diffusion in solids and liquids [24]. Diagrams of the obtained relationships used in the simulation are shown in Figure 1.

### 2.2. Test Casting and Process Parameters

The test casting was a water pump housing produced for one of the world’s leading automotive brands. The castings were made on a Frech DAK720-62 cold chamber die casting machine. The tests were carried out on a 4-cavity die, from which, in order to minimize the impact of non-simultaneous filling of individual cavities, only castings from one precisely defined cavity were selected for further analysis.

Tests included making castings for three selected settings of plunger velocity and also for the three selected values of III-phase pressure settings. Three samples representing wall thicknesses of the casting of 3, 6, and 11 mm, respectively, were cut out. From the samples, specimens for metallographic tests were made. A dose of 1.55 kg was used to produce one bundle of castings shown in Figure 2.

As part of the more extensive research, the following values of the variables marked in Table 2 were selected to analyse the impact of the high-pressure process parameters on the cooling rate and the structure of the 46000 alloy:Test 1: Changing the setting of the compression pressure (process phase III) at a constant velocity of the pressing plunger in phase II of the process (variable data: A, E, and K).Test 2: Changing the velocity setting of the pressing plunger in phase II of the process, at a constant compression pressure (phase III of the process).

A summary of the parameters of the high-pressure die casting process used in the tests is presented in Table 2.

Pump body casting selected for the analysis (which comes from the lower right mould cavity in Figure 2) is shown in Figure 3. From each test casting, three samples representing spots of different wall thickness were taken. Figure 4 shows a CAD drawing of the casting with marked points of sampling for the microstructure analysis as well as fragment of the 3D FEM mesh of the same casting prepared for simulation.
Area 1 (Figure 3a), marked in Figure 3 as “Point3”, with a wall thickness of g = 3 mm.Area 2 (Figure 3a), marked in Figure 3 as “Point4”, with a wall thickness of g = 6 mm.Area 3 (Figure 3b), marked in Figure 3 as “Point5”, with a wall thickness of g = 11 mm.

The remaining operating parameters of the machine and peripheral devices (such as fluid temperature settings in the thermostabilization system; velocity profiles of the first pressing phase; plunger positions at the time of switching movement phases; times of mould opening and part ejection, cooling, and other) remained unchanged and were based on settings that gave a satisfactory quality effect in the production of a casting.

### 2.3. Computer Simulation

Simulations were carried out using Visual-Environment 16.0 (ESI Group, Rungis, France) integrated software with ProCAST 2020.0 solver, ESI Group^®^. Calculations were performed using the finite element method (FEM).

The FEM mesh step size for the casting was taken to be equal to 1 mm (Figure 4b). In the domains of the die bodies, the mesh step was variable, varying from 1 mm on the mould cavity surfaces to 5–8 mm on surfaces most distant from the mould cavity. The average number of surface mesh elements (2D) of one mould cavity walls mapping a casting was about 75,000 elements. The average number of 3D elements for mapping one casting was about 415,000.

A variable value of heat transfer coefficient (HTC) in the contact between the alloy and the forming inserts was assumed, depending on the temperature of the alloy in contact with the mould. HTC = 10 kW/(m^2^·K) was assumed for the liquid state, and 1 kW/(m^2^·K) for the solid state. For the solidification temperature range, the HTC value was approximated by a linear relationship based on the above limits.

To determine the initial die temperature distribution for the die cavity filling analysis, simplified initial calculations of 15 full cycles of the pressure machine were carried out at first. In this series of calculations, in order to reduce simulation time, the analysis of liquid alloy movement phenomena was omitted, and the so-called instantaneous filling of the mould cavity (uniform initial temperature of the alloy in all cavity volume) was assumed. Temperature distribution on the walls of die cavity taken as the initial condition in the cavity filling simulation is shown in Figure 5.

### 2.4. Microstructure and Density

A metallographic characterization was made using a Leica MEF 4M (Leica Microsystems, Wetzlar, Germany) light microscope and a QWin v3.5 (Leica Microsystems, Wetzlar, Germany) image analysis software at various magnifications to examine microstructure of Al-Si-Cu castings. Light microscope examinations included measurements of the fraction of α(Al), f_α_ phase grains, the dendritic cell size d_α_, and the size of silicon particles in eutectics, *L*_max_. Microscopic examinations were carried out on polished and un-etched samples.

All investigated castings were subjected to an average density test. The density of individual samples taken from the casting was measured by hydrostatic weighing (based on Archimedes’ law). Distilled water was used as the working fluid. The RADWAG WAS 160/X laboratory instrument balance with a factory density measurement kit was used.

## 3. Test Results

### 3.1. Alloy Density

The results of the alloy density tests are presented in Table 3. These results make it possible to determine that the increase in the compression pressure increases the average alloy density, while the effect of increasing the velocity of the pressing plunger in the second phase of the process is opposite.

Alloy in the area 1 (Figure 3a) of the tested casting has a local density for the entire range of pressure and filling rate greater than the average density of the entire casting and is located in the most advantageous place as regards liquid alloy feeding. In the area 2 (Figure 3a), alloy has a local density close to the average density of the entire casting obtained in given conditions (pressure, velocity) and has a relatively good liquid alloy feeding. In the area 3 (Figure 3b), alloy shows a local density lower than the average density for the entire range of casting conditions for this casting due to the difficulties with feeding of liquid alloy given the existing solution of the gating system.

Simulations of the die cavity filling, alloy cooling, and solidification were performed to correlate cooling conditions with microstructure test results. The inputs for this simulation were the chemical composition of the alloy, the thermophysical parameters of the materials involved, and the alloy-die and die-environment heat transfer coefficient. In addition to the geometry of the die cavity shown in Figure 2a, the geometry of the complete gating system (pressure chamber, biscuit, and plunger), forming inserts and cores with cooling channels were included in the geometric model.

### 3.2. Cooling Curves

The simulation results of cooling curves and solid fraction changes during solidification obtained by ProCAST software (ESI Group, Rungis, France) are shown in Figure 6. The places for which the curves shown in this figure were recorded correspond to the centre of the casting wall thickness in the positions shown in Figure 3 and Figure 4 and marked “Point3”–“Point5”, respectively.

Figure 7 presents the effects of the influence of wall thickness and plunger velocity *V*_2_ on the cooling rate of the casting close to the equilibrium alloy solidification temperature of α(Al) phase and on the alloy solidification time.

The data presented in Figure 7 show that in the case of a thin-walled casting with wall thickness of 3 mm, the *V*_2_ parameter has the greatest impact on the cooling rate before the solidification process begins. For a casting with wall thickness of 11 mm, the changes in the cooling rate are insignificant. In all analysed cases, entering the scope of thin-walled castings is characterized by an intensive increase in the cooling rate regardless of the value of the *V*_2_ parameter. The highest increase in the cooling rate was observed for the thin-walled casting (*g* = 3 mm) at the highest applied velocity *V*_2_ of 3.5 m/s. The alloy solidification time determined on the basis of the ProCAST simulation in the analysed positions varies from about 1.4 to 6.1 s for castings with a wall thickness of 3 and 11 mm, respectively.

On the basis of the analysis for the tested casting, it is possible to formulate the dependence of the cooling rate *C*_R_ on the local wall thickness (*g*, mm) and plunger velocity during the second phase of pressing (*V*_2_, m/s)—Equation (1) with the correlation coefficient *R*^2^ = 0.99. The dependence of the solidification time *t*_K_ on the wall thickness of the casting and plunger velocity (*V*_2_, m/s) is described by the Equation (2) with the correlation coefficient *R*^2^ = 0.99.
(1)$$C_{R}=-2.36\cdot 10^{4}+2.357\cdot 10^{4}\;e^{\frac{0.035}{g}}+10.46\cdot e^{\frac{2.247}{g*V_{2}}},\;K/s$$
(2)$$t_{K}=1.60+7.538\cdot 10^{-5}\cdot e^{g}-0.003\cdot e^{V_{2}},\;s$$

### 3.3. Microstructures

Figure 8 and Figure 9 show the examples of microstructures that can be found in castings with different wall thicknesses and process parameters. The results of quantitative measurements of the alloy microstructure parameters are presented in a graphical form in Figure 10, Figure 11, Figure 12 and Figure 13. On their basis, it can be seen that increasing wall thickness from 3 to 11 mm has increased the *d*_α_ parameter by an average of 55% in the entire range of the applied pressure *P*_3_ (see Figure 10). Increasing the *P*_3_ pressure in the range of 160–290 bar has an insignificant effect on the reduction of the *d*_α_ parameter in thin walls (approx. 5.5%) and is more visible in the case of thicker casting walls (approx. 7%).

Similarly, the impact of the *V*_2_ velocity settings on the change of the *d*_α_ parameter value in the walls of different thicknesses was assessed. The obtained dependencies are presented in a graphical form in Figure 11.

The analysis of the results (Figure 11) demonstrates that increasing wall thickness from 3 to 11 mm at velocity settings in the range of 0.5 and 0.9 m/s has resulted in more than a twofold increase in the characteristic dimension of the dendritic grain *d*_α_. At *V*_2_ = 3.5 m/s, the influence of wall thickness on this microstructure parameter is not so strong. The *d*_α_ value in an 11 mm wall is only 1.68 times greater than in a 3 mm wall.

Figure 12 shows the results of the assessment of the impact of the *P*_3_ pressure settings on the change in the *L*_max_ parameter value. It can be seen that increasing wall thickness from 3 to 11 mm has resulted in an average about two-fold increase in the *L*_max_ parameter over the entire range of the applied pressure *P*_3_. Moreover, increasing the *P*_3_ pressure in the range of 160–290 bar reduces the *L*_max_ parameter by an average of 22%.

Figure 13 shows the data for assessing the effect of *V*_2_ velocity settings on the change in the maximum silicon particle size *(L*_max_). They show that increasing wall thickness from 3 to 11 mm at the velocity setting of 0.5 m/s has resulted in an almost 2.5-fold increase in the maximum size of silicon particles in the casting wall g = 3 mm, increasing to over 3 times the value for g = 11 mm and velocity *V*_2_ = 3.5 m/s. A decreasing effect of increasing the velocity on *L*_max_, reaching at *V*_2_ = 3.5 m/s the value of *L*_max_ = 22 µm can be observed. Additionally, it can be concluded that increasing the velocity of *V*_2_ at a given pressure results in a more pronounced decrease in *L*_max_ in the wall g = 3 mm (by about 40%), by a half lower in the case of g = 6 mm (by about 25%), and the smallest decrease in the wall g = 11 mm (about 20%).

Increasing wall thickness of the castings significantly affects the cooling rate (Figure 7a) and the resulting solidification time (Figure 7b). The analysis of Figure 10, Figure 11, Figure 12 and Figure 13 demonstrates that the change in wall thickness of the castings in the entire tested range, i.e., from 3 mm to 11 mm, fundamentally changes the structure parameters. Examples of the microstructure of the samples taken from castings obtained in a wide range of process parameters and for various wall thicknesses shown in Figure 8 and Figure 9 do not exhibit the dendritic morphology typical of Al-Si alloys with visible primary and secondary branches. The microstructure of the tested castings shows a dendritic-cellular character. A comparison of the microstructure of samples taken from castings obtained over a wide range of process parameters (Figure 8 and Figure 9) shows that, with decreasing wall thickness and increasing cooling rate, a transition is observed from typical dendritic grain morphology to the dendritic seaweed type morphology, as is observed in different alloys under the condition of forced cooling [25,26].

The castings’ microstructure (Figure 14) consists essentially of dendritic cells of the α (Al) phase, eutectic α (Al) + Si, iron-reach phases α-Al_15_(Fe, Mn, Cr)_3_Si_2_ and the phases typical of the 46000 alloy, i.e., Mg_2_Si and Al_8_Mg_3_FeSi_6_ (Q phase). The silicon eutectic contains silicon plates, the size of which increases by increasing the wall thickness of the castings. As can be seen in Figure 8 and Figure 9, the eutectic in the thin-walled, 3 mm thick part is fine and non-uniformly distributed. It occupies large areas in interdendritic spaces. As the wall thickness increases, the lamellar eutectic takes a more uniform distribution.

Changing wall thickness and process parameters significantly affects the size of dendritic cells. The size of dendritic cells grows together with the increase of the wall thickness of the castings. From Figure 10, it can be seen that increasing wall thickness from 3 to 11 mm exponentially increases the size of the dendritic cells from about 12 to 18–20 µm. It can be noticed that increasing the compression pressure in the analysed range slightly reduces their size. A much greater effect on the size of dendritic cells was observed for the analysed range of changes in the plunger velocity in the second phase of the HPDC process cycle. Increasing plunger velocity significantly reduces the size of dendritic cells, and this effect is greater for thicker parts of the castings. The dependence of the size of eutectic silicon plates on the wall thickness of the castings and the process parameters, i.e., compression pressure and plunger velocity, is qualitatively convergent with their influence on the size of dendritic cells.

The Sludge Factor (SF) parameter of Al-Si alloys, which is expressed as [21], was determined for the experimental alloy:(3)SF=Fe+2Mn+3Cr
where Fe, Mn, and Cr are wt. percent of iron, manganese, and chromium, respectively.

For the analysed alloy, SF = 1.799, which corresponds to the proportion of Fe-containing particles at the level of 1% vol. The research [22] demonstrates that for the tested alloy, increasing the SF to the level of 3.7 increases the proportion of iron-reach phases to the level of 3% vol. It is generally accepted that an Fe:Mn ratio of less than 2 is necessary to prevent the formation of acicular β-Fe [22]. Fe:Mn ratio in the present study is equal to 2.04, so it is just above this limit. However, the presence of acicular β-Fe was not observed. The presence of coarse intermetallic compounds such as the α-Al_x_(Fe,Mn,Cr)_y_ sludge particles can significantly affect the crack nucleation stage, which has a clear negative effect on the reduction of the fatigue strength of the alloys, as reported in [21].

Changes of the phase composition of the investigated alloy were calculated using “Thermo-Calc 2019a” database TCAL7. Results of the Thermo-Calc calculation for the chemical composition of the investigated alloy presented in Table 1 are shown in Figure 15. Figure 15a shows changes in the equilibrium phase composition of the analysed alloy occurring with changes in temperature. A section of this graph corresponding to the range of the intermetallic phase’s precipitation is extracted in Figure 15b. Hence, it follows that at ambient temperature in thermodynamic equilibrium conditions the microstructure should mainly consist of a α(Al) (FCC_L12), Si (DIAMOND_A4) and a number of intermetallic phases inclusions: Al_15_Si_2_M_4_, Al_9_Fe_2_Si_2_, Al_18_Fe_2_Mg_7_Si_10_, Al_8_Mg_3_FeSi_6_, and Al_2_Cu_C_16_.

According to the thermodynamic database, at the temperature of the analysed alloy in the reheating furnace (750 °C) in the bath, apart from the liquid phase, the precipitation of the Ti_7_Al_5_Si_14_ intermetallic phase should occur. When lowering the alloy temperature to the temperature of the initiation of α-Al solidification, particles of other intermetallic phases may solidify from the liquid, i.e., Al_13_M_4_ and Al_15_(Fe,Mn)_3_Si_2_, called *sludge particles* in the literature. The name is related to the fact that the presence and increase in the amount of these precipitates increases the viscosity of the suspension. The solidification of α-Al dendrites begins after exceeding 590.9 °C and the solidification of the α-Al + Si eutectic after exceeding 566.8 °C. In equilibrium conditions, solidification is completed at a temperature of approximately 478°C. In the course of further cooling in the solid state, particles of other intermetallic phases (β-AlFeSi, Al_9_FeNi, Q phase—Al_5_Cu_2_Mg_8_Si_6_, θ-AlCu, and Al_3_Ni phase) may be precipitate. These particles are called secondary precipitations.

As it follows from Figure 8, Figure 9 and Figure 14, the average size of the primary particles (sludge particles), which solidify directly from the liquid phase, is 18–35 µm. The size of the secondary particles is 3–5 µm.

The analysis of the microstructure showed that reducing wall thickness, i.e., increasing the cooling rate in the analysed range, causes an insignificant reduction in the size of the primary iron-reach particles. Their largest fraction occurred in castings obtained with the lowest compression pressure (castings from the A series). Increasing the compression pressure significantly affects the fragmentation of the primary iron-reach particles. This is already noticeable with a compression pressure of 210 bar (samples from the E series). The role of plunger velocity in the second phase of the cycle is analogous to the effect of compression pressure *P*_3_. For the lowest value of plunger velocity, the highest share of primary iron-reach particles was observed. Together with the increase of plunger velocity, their fraction and size decrease. The cooling rate thus affects the sludge particles which is in line with the research carried out by Ferraro, Ceschini [20,27].

In addition to the primary particles (sludge particles), there are secondary, fine polyhedral α-Fe particles in the microstructures, as shown in Figure 8 and Figure 9. These intermetallic were detected in all the experimental alloys and are distributed at the interdendritic regions. Due to their small size, secondary particles tend to form clusters of a few to a dozen or so particles. It is known [20,27] that the particle size is of key importance for the process of their absorption/repulsion by the solidification front. The smaller size of the particles, the greater the necessary cooling rate that allows them to be absorbed by the solidification front, which leads to a greater homogeneity of their distribution in the alloy microstructure. It is worth noting that the use of compression pressure at the upper level of 290 bar and the higher plunger velocity in the second phase of the HPDC process cycle has reduced but not completely eliminated the formation of secondary particles.

## 4. Summary and Conclusions

The presented studies not only confirm the general literature data, obtained mainly as a result of simulation calculations, on the effect of wall thickness of a high-pressure die casting on the solidification rate, but also expand them. A significant differentiation of the microstructure and phase composition of the EN-AC 46000 aluminium-based alloy was demonstrated, occurring in different regions of the casting obtained in the same process conditions.

On the basis of the conducted research, the following conclusions can be drawn.
The analysed high-pressure casting with various wall thicknesses (3–11 mm) was characterized by a high variability of the cooling rate and solidification time. The highest cooling rate was observed in the casting with the least analysed wall thickness of g = 3 mm at the highest plunger velocity in the second phase of the HPDC *V*_2_ cycle, amounting to 3.5 m/s. The solidification time of the alloy determined on the basis of the ProCAST simulation at the analysed points varied from about 1.4 s for a wall thickness of 3 mm to 6.1 s for a wall thickness of 11 mm.The mathematical dependencies linking wall thickness of the casting, plunger velocity, and cooling rate, as well as wall thickness, compression pressure, and solidification time, were presented.Different microstructures of EN 46000 alloy were obtained for castings with wall thicknesses in the range of 3 up to 11 mm. The microstructures were characterized in detail, quantifying the average diameter of dendritic cells, maximum size of silicon particles, as well as size and distribution of iron-reach particles. These features became finer as the solidification rate increases. It was shown that increasing compression pressure insignificantly reduced the size of dendritic cells. Reducing the filling time of the die cavity by increasing plunger velocity had a much greater effect on reducing the dimensions of dendritic cells. This impact was greater for thicker parts of the castings.The analysis of the microstructure showed that reducing wall thickness, i.e., increasing cooling rate, caused a slight reduction in the size of the primary iron-reach particles. Their largest share occurred in castings obtained with the lowest compression pressure (castings from the A series). Increasing compression pressure significantly affected the fragmentation of the primary iron-reach particles. It was already noticeable with a compression pressure of 210 bar (samples from the E series). The role of the plunger velocity in the second phase of the cycle was analogous to the effect of the compression pressure *P*_3_. For the lowest value of plunger velocity, the highest share of primary iron-reach particles was observed.

## Figures and Tables

**Figure 1 materials-15-05702-f001:**
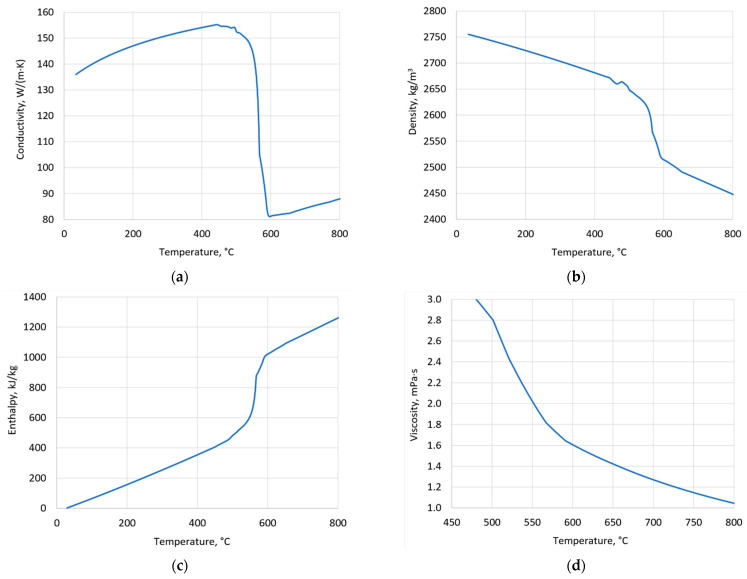
Temperature dependencies of thermophysical parameters of alloy (CompuTherm LLC Thermodynamic Databases): conductivity (**a**); density (**b**); enthalpy (**c**); viscosity (**d**).

**Figure 2 materials-15-05702-f002:**
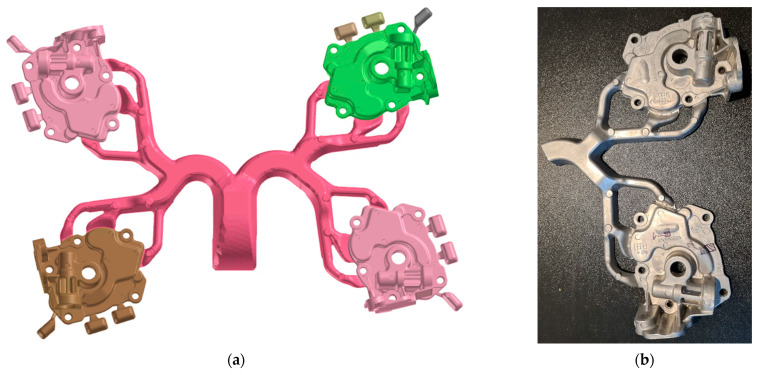
View of the bundle: geometric model (ProCAST) of the bundle made in a 4-cavity die used in the simulation (**a**) and an experimental part consisting of two test castings (**b**).

**Figure 3 materials-15-05702-f003:**
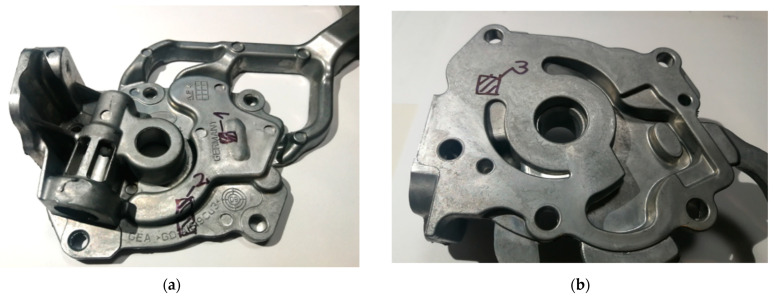
Sampling locations for metallographic tests and microstructure analysis: view from the side of the movable die (**a**); view from the side of the stationary die (**b**).

**Figure 4 materials-15-05702-f004:**
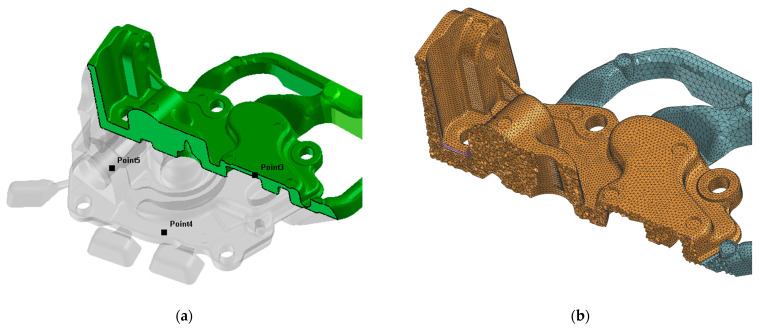
3D geometry of the casting with marked places of analysis in locations as in Figure 3 (**a**) and fragment of ProCAST 3D FEM mesh for the same part of the casting (**b**).

**Figure 5 materials-15-05702-f005:**
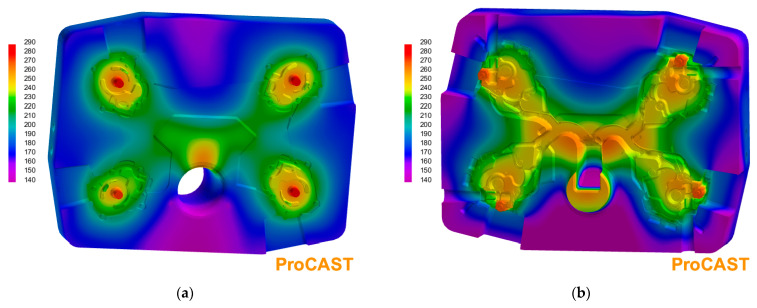
Temperature distribution on surfaces of forming inserts used as initial condition for simulation of die cavity filling: fixed die (**a**) and mobile die (**b**).

**Figure 6 materials-15-05702-f006:**
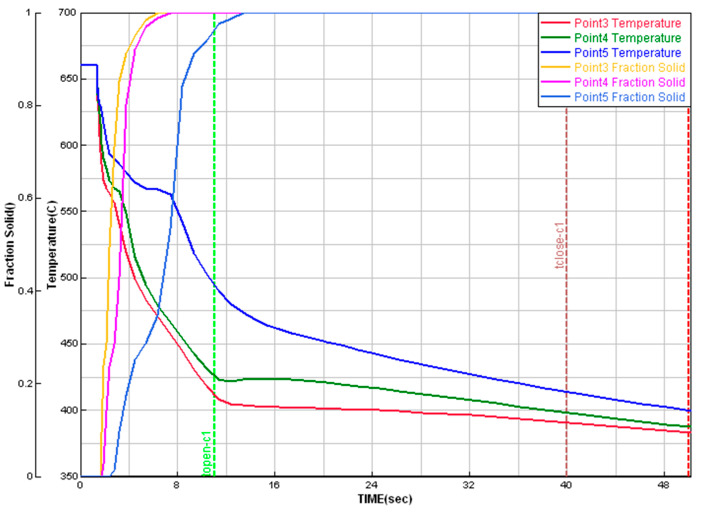
Simulated cooling curves and fraction solids in samples with different wall thicknesses (ProCAST).

**Figure 7 materials-15-05702-f007:**
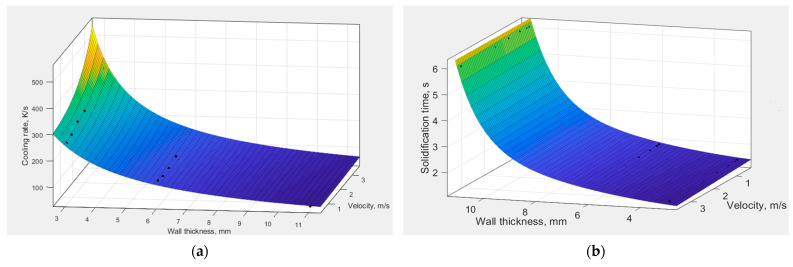
Influence of wall thickness and plunger velocity on cooling rate (**a**) and alloy solidification time (**b**).

**Figure 8 materials-15-05702-f008:**
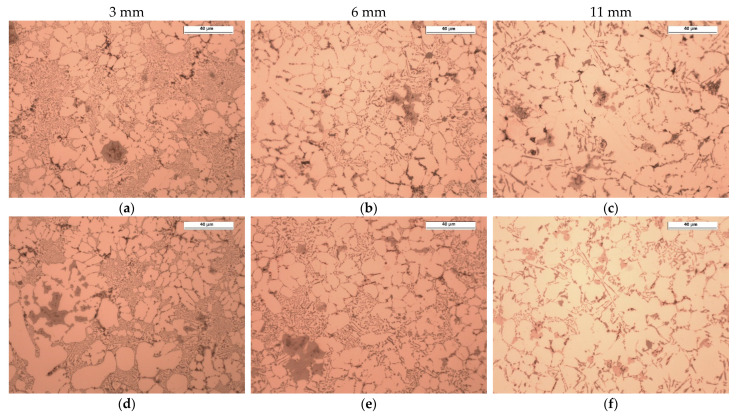
Microstructure of the tested Al-Si-Cu alloy in areas with different wall thicknesses in test castings according to Table 3: series 1 (**a**–**c**), series 5 (**d**–**f**), and series 15 (**g**–**i**); unetched.

**Figure 9 materials-15-05702-f009:**
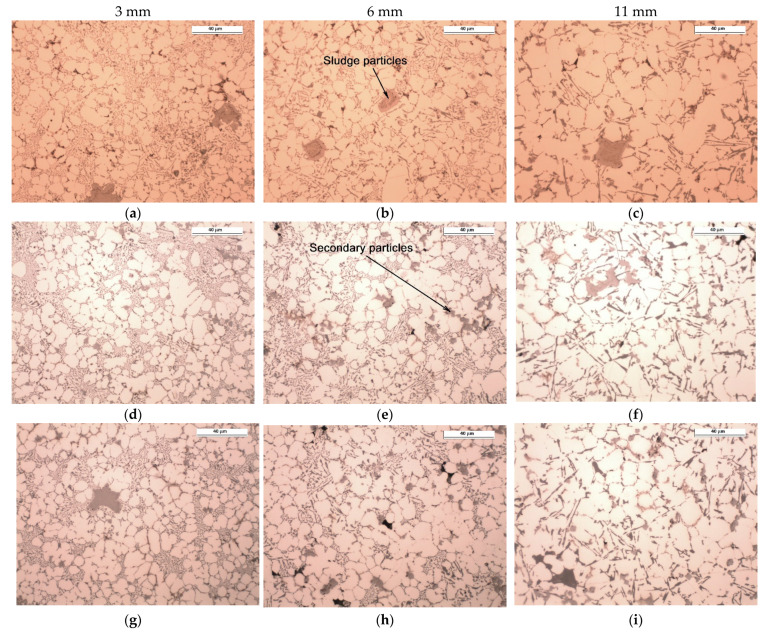
Microstructure of the tested Al-Si-Cu alloy in areas with different wall thicknesses in test castings according to Table 3: series A (**a**–**c**), series E (**d**–**f**), and series K (**g**–**i**); unetched.

**Figure 10 materials-15-05702-f010:**
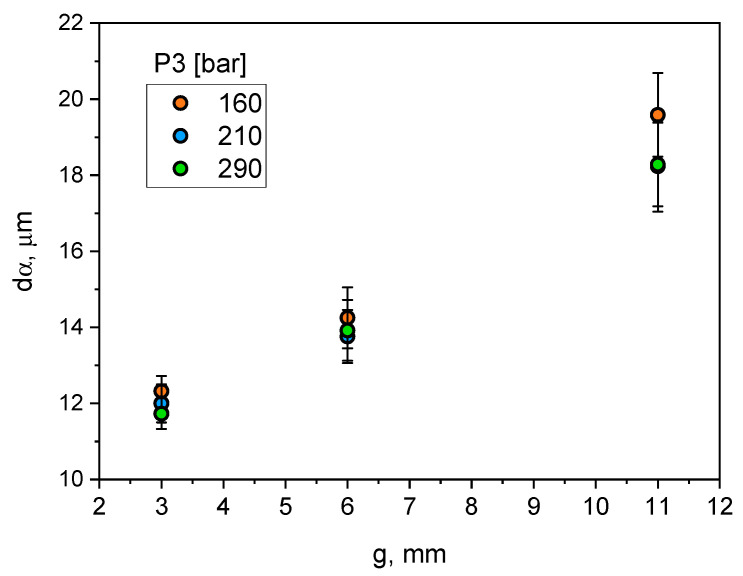
Influence of casting wall thickness and compression pressure on *d*_α_ parameter.

**Figure 11 materials-15-05702-f011:**
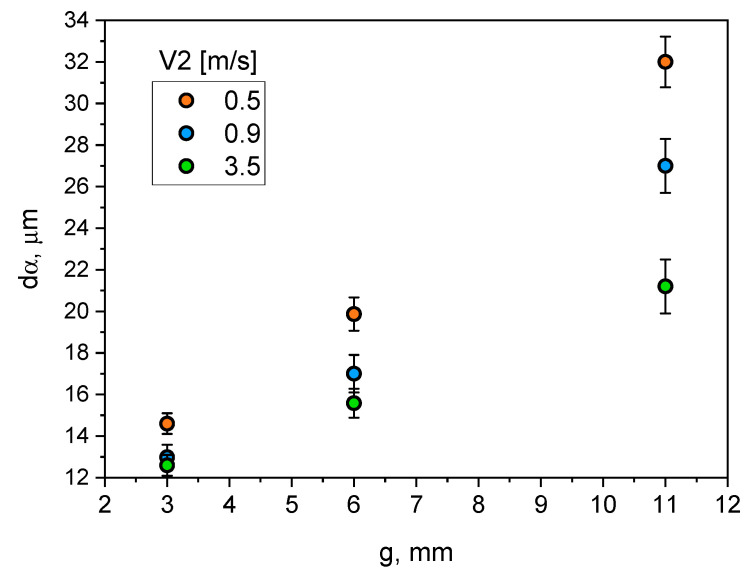
Influence of casting wall thickness and plunger velocity in the pressing chamber on *d*_α_ parameter.

**Figure 12 materials-15-05702-f012:**
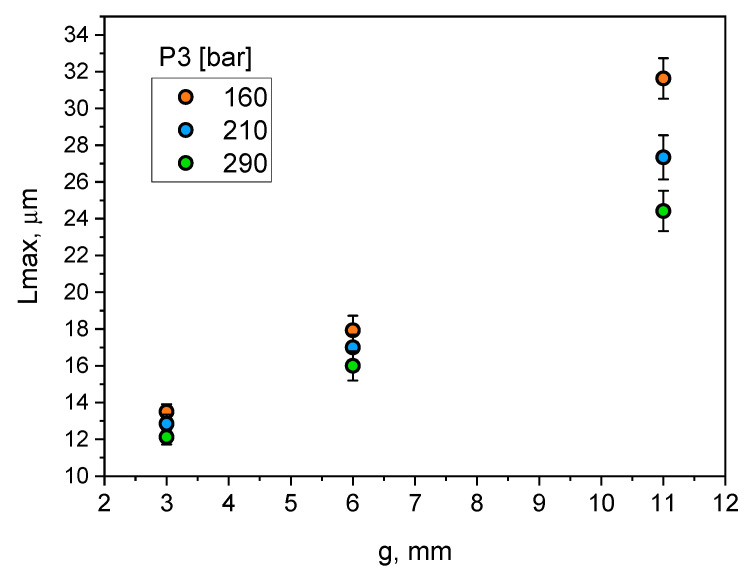
Influence of casting wall thickness and compression pressure on *L*_max_ parameter.

**Figure 13 materials-15-05702-f013:**
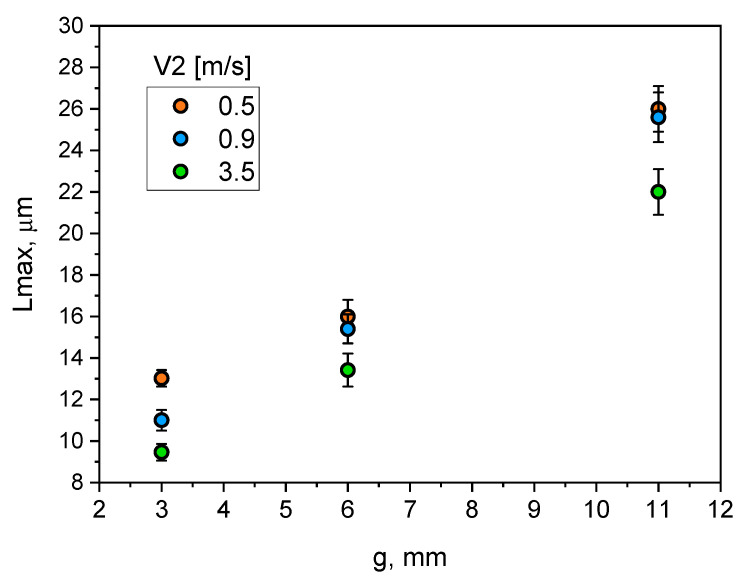
Influence of wall thickness and plunger velocity (*V*_2_) on maximum size of silicon particles (*L*_max_).

**Figure 14 materials-15-05702-f014:**
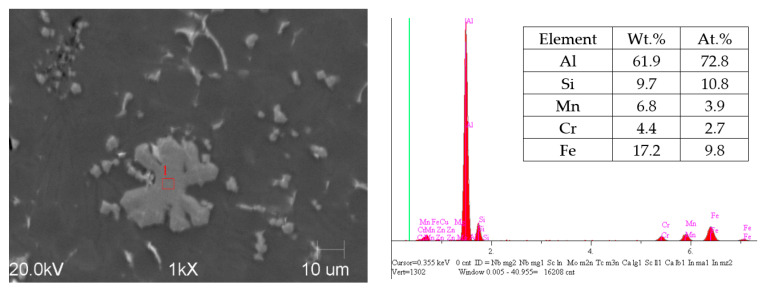
Microstructure of tested alloy in the casting with M series (Table 1)—area of wall thickness 6 mm with EDS analysis of iron-rich particle.

**Figure 15 materials-15-05702-f015:**
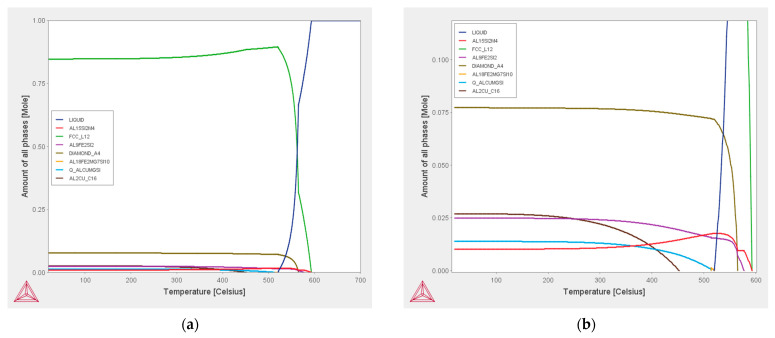
Thermo-Calc analysis of fraction of all phases formed in analysed alloy (equilibrium condition): full range (**a**) and magnification for range of intermetallic phases content (**b**).

**Table 1 materials-15-05702-t001:** Chemical composition of the experimental alloys.

No.	Element	Content, Mass %
1	Si	8.782
2	Fe	0.775
3	Cu	2.280
4	Mn	0.380
5	Mg	0.462
6	Cr	0.088
7	Zn	0.880
8	Ni	0.033
9	Ti	0.050
10	Pb	0.053
11	Sn	0.013
12	Sb	<0.005
13	Al	86.201

**Table 2 materials-15-05702-t002:** Process parameters for obtaining selected test castings.

Series	TEST 1: *V*_2_ = 3.5 m/s; *t*_2_ = 34 ms; *V*_gate_ = 31.20 m/s		
	*P* _3set_	*P* _3meas_	*P* _spec_
	[MPa]	[MPa]	[MPa]
A	160	158	550
B	180	179	615
E	210	208	717
J	260	257	886
K	270	267	921
M	290	286	986
Series	TEST 2: *P*_3_ = 240 MPa; *P*_spec_ = 820 MPa		
	*V* _2_	*t* _2meas_	*V* _gate_
	[m/s]	[ms]	[m/s]
1	0.5	235	5.25
2	0.6	196	5.55
5	0.9	129	8.28
10	1.4	83	12.67
12	2.0	78	17.74
14	3.0	47	26.60
15	3.5	35	31.20

*V*_2_—plunger velocity in an II phase; *V*_gate_—metal velocity in gates; *t*_2_—filling time set. *t*_2meas—_measured filling time; *P*_3set_—intensification pressure set; *P*_3meas_—intensification pressure measured; *P*_spec_—specific casting pressure; *t*_2_—filling time; *t*_2meas_—measured filling time.

**Table 3 materials-15-05702-t003:** Results of the average casting density tests.

Series	TEST 1: *V*_2_ = 3.5 m/s; *t*_2_ = 34 ms; *V*_gate_ = 31.20 m/s	
	*P* _3set_	Average Density of Test Casting, *D*_av_
	[MPa]	[g/cm^3^]
A	160	2.680
E	210	2.703
K	290	2.711
	TEST 2: *P*_3_ = 240 MPa; *P*_spec_ = 820 MPa	
	*V* _2_	Average density of test casting, *D*_av_
	[m/s]	[g/cm^3^]
1	0.5	2.727
5	0.9	2.722
15	3.5	2.687

## Data Availability

Not applicable.

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
