# Peer review of "Influence of High-Pressure Die Casting Parameters on the Cooling Rate and the Structure of EN-AC 46000 Alloy"

_materials, 2022, doi:10.3390/ma15165702_

Round 1

Reviewer 1 Report

In this work, the effects of the parameters of the high-pressure die casting process, mainly the pressure as well as the rate and time of filling the die, on the parameters of the casting microstructure in thin and thicker sections located closer to or further from the gating were systematically investigated. The work of the paper is well-done, with some interesting results. However, it requires an appropriate revision before publication.

1. More details of the experiment should be provided, and the simulated thermal physical parameters should also be provided.

2. The rulers in Fig. 7 and 8 should be improved.

3. Table 3, “Mpa” should be “MPa”.

4. High magnification microstructures of the tested Al-Si-Cu alloy with different wall thicknesses should be provided in order to better show the microstructure changes under different conditions.

5. More theoretical analyses of the correlation of the casting parameters and microstructure of the aluminum alloys are welcome. Adding follow references enrich your introduction and discussion, providing more information for your work. (1) Combined effects of mechanical vibration and wall thickness on microstructure and mechanical properties of A356 aluminum alloy produced by expendable pattern shell casting. Materials Science and Engineering A, 2014, 619(12): 228-237.  (2) Correlation of microstructure with mechanical properties and fracture behavior of A356-T6 aluminum alloy fabricated by expendable pattern shell casting with vacuum and low-pressure, gravity casting and lost foam casting. Materials Science and Engineering A, 2013, 560(1): 396-403.  

6. The conclusion is usually in the past tense.

Author Response

Respond to Reviewer 1

Thank you very match for your analysis and comments of our manuscript. We confirm that their inclusion has improved the quality of our paper.

In accordance with the comments of all reviewers, the order of some parts of the text has been changed compared to the 1st reviewed version. For this reason, a different subsection division of the manuscript was introduced. Several additional figures and literature items have also been added.

In the case of Comments 4, we decided to leave the images obtained at 500x magnification, which is justified below.

With regard to your comments, we kindly inform you as follows:

Ad 1. More details of the experiment should be provided, and the simulated thermal physical parameters should also be provided.

Section “2. Description of the material and experiments” is extended and reordered. Estimated temperature dependences of the thermal physical parameters are presented in new Figure 1 (line 131).

Ad 2. The rulers in Fig. 7 and 8 should be improved

Images with better resolution are used in Fig. 8 and 9 (former Fig. 7 and 8) ­– pp. 13 and 14.

Ad 3. Table 3, “Mpa” should be “MPa”.

This error was corrected (line 229).

Ad 4. High magnification microstructures of the tested Al-Si-Cu alloy with different wall thicknesses should be provided in order to better show the microstructure changes under different conditions.

Metallographic analysis was performed over a wide range of magnifications (from 25x to 1000x). In this paper, images taken at 500x magnification were chosen for Figures 8 and 9 because of the following reasons:

- eutectic morphology (lamellar, fibrous) is visible;

- at least a few dozen dendritic grains are visible;

- the presence of secondary dendrite arms is also evident.

Ad 5. More theoretical analyses of the correlation of the casting parameters and microstructure of the aluminum alloys are welcome. Adding follow references enrich your introduction and discussion, providing more information for your work. …

(1) Combined effects of mechanical vibration and wall thickness on microstructure and mechanical properties of A356 aluminum alloy produced by expendable pattern shell casting. Materials Science and Engineering A, 2014, 619(12): 228-237.

(2) Correlation of microstructure with mechanical properties and fracture behavior of A356-T6 aluminum alloy fabricated by expendable pattern shell casting with vacuum and low-pressure, gravity casting and lost foam casting. Materials Science and Engineering A, 2013, 560(1): 396-403.

Thank you. This comment has been reflected in the manuscript. Above position are cited in lines 92-96 of the revised version of the manuscript as [21,22].

Ad 6. The conclusion is usually in the past tense.

Text of the conclusion has been corrected.

Reviewer 2 Report

The authors studied in detail the effect of die-casting process parameters on the microstructure of the castings. This study has a very strong engineering application background and provides a reliable experimental basis for the production of aluminum alloy die castings. We hope that the authors will improve and refine the following issues.

1. The authors used Procast to simulate the process of die casting and obtained the corresponding results of the solidification process of the castings. Nevertheless, the authors should give detailed parameters for the Procast setup, including mesh size, boundary conditions, simulation parameters, etc.

2. The authors did not establish a relationship between the die casting process parameters and the mechanical properties. If possible, it is recommended to take tensile specimens at different wall thickness parts of the casting.

3. The author notes that Figs. 7 and 8 do not exhibit the dendritic morphology typical of Al-Si alloys. This is imprecise descriptions, and in fact the some dendrite structure can still be seen clearly in these images.

4. Finally, how the different process parameters affect the die casting defects. In addition, images of the simulation results of the solidification process with different process parameters are very much recommended to be put in the supporting information, they are very important results for the numerical simulation of the casting process.

Author Response

Respond to Reviewer 2

Thank you very match for your analysis and comments of our manuscript. We confirm that their inclusion has improved the quality of our paper.

In accordance with the comments of all reviewers, the order of some parts of the text has been changed compared to the 1st reviewed version. For this reason, a different subsection division of the manuscript was introduced. Several additional figures and literature items have also been added.

With regard to your comments, we kindly inform you as follows:

  1. The authors used Procast to simulate the process of die casting and obtained the corresponding results of the solidification process of the castings. Nevertheless, the authors should give detailed parameters for the Procast setup, including mesh size, boundary conditions, simulation parameters, etc.

The comment has been reflected in the manuscript. Thermal physical parameters of the alloy are presented in new Figure 1 (line 131). New Figure 4b (line 179) and Figure 5 (line 208) and new subsection “2.3. Computer simulation” are added (lines 186-209) of the revised version of the manuscript).

  1. The authors did not establish a relationship between the die casting process parameters and the mechanical properties. If possible, it is recommended to take tensile specimens at different wall thickness parts of the casting

Due to the geometry and size of the analyzed castings, it was not possible to prepare specimens for testing the strength properties. It is planned to carry out these tests using a new instrumentation according to the concept described in the article A.R. Anilchandra at al. Evaluating the Tensile Properties of Aluminum Foundry Alloys through Reference Castings - A Review. Materials 2017, 10, p. 1011. doi:10.3390/ma10091011.

  1. The author notes that Figs. 7 and 8 do not exhibit the dendritic morphology typical of Al-Si alloys. This is imprecise descriptions, and in fact some dendrite structure can still be seen clearly in these images.

We agree with your comment. Consequently, the following text has been included in lines 328-333 and references 25 and 26 have been added.

“A comparison of the microstructure of samples taken from castings obtained over a wide range of process parameters (Figs. 8 and 9 – former 7 and 8) shows that, with decreasing wall thickness and increasing cooling rate results in transition from typical dendritic morphology to the dendritic seaweed type morphology, like observed in different alloys under the condition of forced cooling [25,26].”

[25] Castle, E.G.; Mullis, A.M.; Cochrane, R.F. Evidence for an extensive, undercooling-mediated transition in growth orientation, and novel dendritic seaweed microstructures in Cu–8.9 wt.% Ni. Acta Materialia 2014, 66, pp. 378–387. http://dx.doi.org/10.1016/j.actamat.2013.11.027

[26] Haque, N.; Mullis, A.M. Existence of seaweed structures in rapidly solidified Ni3Ge intermetallic. Journal of Alloys and Compounds 2019, 801, pp. 640-644. https://doi.org/10.1016/j.jallcom.2019.06.050

  1. Finally, how the different process parameters affect the die casting defects. In addition, images of the simulation results of the solidification process with different process parameters are very much recommended to be put in the supporting information, they are very important results for the numerical simulation of the casting process.

The authors have begun preparing material for inclusion in the supporting information for the article. It is planned to develop videos showing the processes of filling the entire mould cavity under the pressure of the pressure machine piston and enlargement for the cavity of the analysed casting, as well as the sequence of solidification of the alloy in the analysed casting.

Reviewer 3 Report

1. Please use "solidification" term instead of "crystallization".

2. How was the density of castings determined? Please add this information to "Description of materials and experiments" section.

3. Information about metallographic characterization, ProCast simulation and Thermo-Calc calculations must be in "Description of materials and experiments" section. This is not the results of the work.

4. Please add the references to the used for ProCast simulation interface heat transfer coefficients.

5. Page 9, line 201: Please replace "Phase analysis" to "Phase composition". Phase analysis cannot be calculated.

6. Page 9, Fig. 4: The content of the phases with low fractions is not seen in the figure. Maybe the logarithmic scale is better in this case, or insert with enlarged part of the figure.

7. Fig. 7 and 8: The great change in eutectic fraction for different wall thickness is clearly seen. A discussion of this effect is needed.

Author Response

Respond to Reviewer 3

Thank you very match for your analysis and comments of our manuscript. We confirm that their inclusion has improved the quality of our paper.

In accordance with the comments of all reviewers, the order of some parts of the text has been changed compared to the 1st reviewed version. For this reason, a different subsection division of the manuscript was introduced. Several additional figures and literature items have also been added.

With regard to your comments, we kindly inform you as follows:

  1. Please use "solidification" term instead of "crystallization".

Corrected in the whole text.

  1. How was the density of castings determined? Please add this information to "Description of materials and experiments" section.

Density was measured by hydrostatic weighing. Proper information is added in lines 218-221.

  1. Information about metallographic characterization, ProCast simulation and Thermo-Calc calculations must be in "Description of materials and experiments" section. This is not the results of the work.

In line with this comment, the content of the article and the division into subsections have been reordered.

However, the information on predicting the phase composition of the alloy on the basis of thermodynamic calculations has been combined with the description of the results of the metallographic study of the phase composition (lines 369-394), as, in the authors' opinion, this forms a logical whole.

  1. Please add the references to the used for ProCast simulation interface heat transfer coefficients.

The description of the heat transfer coefficient at the alloy-die interface used in the calculations has been added in lines 196-200. The HTC relationship used as a function of alloy temperature is recommended by the software manufacturer.

  1. Page 9, line 201: Please replace "Phase analysis" to "Phase composition". Phase analysis cannot be calculated.

This is corrected (lines 369, 372 and 421).

  1. Page 9, Fig. 4: The content of the phases with low fractions is not seen in the figure. Maybe the logarithmic scale is better in this case, or insert with enlarged part of the figure.

As the phase fraction diagrams pass through a point with an ordinate equal to zero, instead of using a logarithmic scale, figure b was added (line 378). The magnification of the ordinate axis in this picture is adjusted for the range of intermetallic phase fraction.

  1. Fig. 7 and 8: The great change in eutectic fraction for different wall thickness is clearly seen. A discussion of this effect is needed

This thread was omitted in the article on purpose. The basis for the discussion of this effect should be the results of fractions measurements of the eutectic colonies and dendrites of α(Al).

Trials of such measurements were made. It turned out that the measurements were subject to a large error. At this stage of the research, their publication was abandoned. The authors are working on ways to provide better contrast of components during microstructure characterization to increase the accuracy of measurements and plan to obtain such results in future studies.

Round 2

Reviewer 1 Report

The paper has been satisfactorily revised on the basis of the proposed comments, and its quality has been significantly improved in the revised paper. Therefore, it is recommended for publication in its present form.

Reviewer 2 Report

The manuscript has been sufficiently improved to warrant publication in Materials